# Proteomics and constraint-based modelling reveal enzyme kinetic properties of *Chlamydomonas reinhardtii* on a genome scale

Marius Arend [1,2,3], David Zimmer[4], Rudan Xu[1,2], Frederik Sommer [5], Timo Mühlhaus[4] & Zoran Nikoloski [1,2,3] ✉

Metabolic engineering of microalgae offers a promising solution for sustainable biofuel production, and rational design of engineering strategies can be improved by employing metabolic models that integrate enzyme turnover numbers. However, the coverage of turnover numbers for *Chlamydomonas reinhardtii*, a model eukaryotic microalga accessible to metabolic engineering, is 17-fold smaller compared to the heterotrophic cell factory *Saccharomyces cerevisiae*. Here we generate quantitative protein abundance data of *Chlamydomonas* covering 2337 to 3708 proteins in various growth conditions to estimate in vivo maximum apparent turnover numbers. Using constrained-based modeling we provide proxies for in vivo turnover numbers of 568 reactions, representing a 10-fold increase over the in vitro data for *Chlamydomonas*. Integration of the in vivo estimates instead of in vitro values in a metabolic model of *Chlamydomonas* improved the accuracy of enzyme usage predictions. Our results help in extending the knowledge on uncharacterized enzymes and improve biotechnological applications of *Chlamydomonas*.

Microalgae can synthesize a wide range of high-value compounds[1] and biofuel precursors[2,3] using industrial waste products and light energy, rendering them a key biotechnological resource propelling the transition to a net-zero carbon economy[4]. However, the economic feasibility of photosynthetic bioreactors requires further optimization of desired biotechnological objectives[4], including the production of lipids[5,6], pigments[7], or hydrogen[8]. Our ability to rationally engineer metabolism for biotechnological applications scales with our understanding of the metabolism of organisms used as cell factories. Genome-scale metabolic models (GEMs), as mathematical representations of knowledge about metabolism, along with constraint-based modeling, have facilitated the design of metabolic engineering strategies[9]. Moreover, enzyme-related constraints that rely on turnover numbers ($k_{cat}$) have been shown to accurately predict various phenotypes, including overflow metabolism[10–12], even without the

usage of measurements of uptake fluxes[12]. Further, these protein-constrained GEMs (pcGEMs) have been used to successfully identify engineering targets for biotechnological applications, such as increased production of lysine[13] or of high-value compounds[14,15] in *Escherichia coli* as well as the increase of the heme protein yield in *Saccharomyces cerevisiae* by 70-fold[16].

The $k_{cat}$ data used in most pcGEM studies are obtained by laborious purification of the enzyme of interest and quantifying its maximum catalytic efficiency in an in vitro experiment[12]. For organisms with available quantitative proteomic and physiological data, it is also possible to estimate the maximal apparent catalytic rate ($k_{app}^{max}$) of an enzyme in vivo using constraint-based modeling[12]. To this end, most studies used parsimonious flux balance analysis (pFBA)[17,18] to obtain estimates of each intracellular flux and find their ratio to the abundance of the corresponding enzymes; this results in condition-specific

[1]Bioinformatics, Institute of Biochemistry and Biology, University of Potsdam, 14476 Potsdam, Germany. [2]Systems Biology and Mathematical Modelling, Max Planck Institute of Molecular Plant Physiology, 14476 Potsdam, Germany. [3]Bioinformatics and Mathematical Modeling Department, Center of Plant Systems Biology and Biotechnology, 4000 Plovdiv, Bulgaria. [4]Computational Systems Biology, TU Kaiserslautern, 67663 Kaiserslautern, Germany. [5]Molecular Biotechnology & Systems Biology, TU Kaiserslautern, 67663 Kaiserslautern, Germany. ✉e-mail: nikoloski@mpimp-golm.mpg.de

$k_{app}$ values which are subsequently compared over the investigated experimental conditions to identify the largest. The latter is then used as the maximal in vivo catalytic rate for the enzymatic reaction, $k_{app}^{max}$. Alternatively, fitting of $k_{cat}$ in metabolic models that explicitly describe protein biosynthesis has also been employed as a strategy for parameter estimation[19,20].

While it has been shown that $k_{cat}$ and $k_{app}^{max}$ values for *E. coli* show high concordance[17], for eukaryotic organisms like *S. cerevisiae*[18] and *Arabidopsis thaliana*[21] lower correlation values between $k_{cat}$ and $k_{app}^{max}$ have been reported. This raises the question about the extent to which in vitro data can describe in vivo enzyme properties, particularly in eukaryotes. Nevertheless, most pcGEMs constructed to date rely on turnover numbers compiled in the public databases, such as: BRENDA[22] and SABIO-RK[23]. While these databases offer comprehensive kinetic data for *E. coli*, only 10% of the entries in the union of the two databases cover enzymes of the Viridiplantae taxon. Further, the databases contain a total of only 85 turnover numbers (0.0012% of entries) specific to green algae. Thus, to render the powerful pcGEM modeling framework applicable with organisms from this biotechnologically relevant taxon, we must substantially increase the knowledge of organism- and taxon-specific turnover numbers.

Here we used cutting-edge mass spectrometry techniques[24,25] to acquire a comprehensive set of protein abundance values from cultures of *Chlamydomonas reinhardtii* wild type and mutant strains grown under various conditions. We used this data set together with the recently developed minimization of non-idle enzyme (NIDLE) approach[26] to estimate $k_{app}^{max}$ values for reactions catalyzed by single enzymes as well as decomposing the contribution of isoenzymes to their catalyzed reactions, thus extending the state-of-the-art for estimation of $k_{app}^{max}$ values by constraint-based modeling. Due to these improvements we achieved a higher $k_{app}^{max}$ coverage (24% of enzymatic reactions) than previous works[17,18,21], extending the available literature data on *C. reinhardtii* by ~ 10-fold. In total, we obtained $k_{app}^{max}$ values for 568 reactions including 46 transport reactions whose transport capacities are notoriously difficult quantify with current in vitro techniques. Our subsequent analysis corroborated the low correspondence between $k_{cat}$ and $k_{app}^{max}$ values in eukaryotic organisms. In line with these results, we showed that the substitution of $k_{cat}$ values in pcGEMs of *C. reinhardtii* with $k_{app}^{max}$ estimates improved predictive accuracy of enzyme resource allocation.

## Results and discussion

### High-quality protein abundance data from various experimental set-ups enable $k_{app}^{max}$ estimation

To obtain $k_{app}^{max}$ values for *C. reinhardtii*, we generated a comprehensive, high-quality proteomics data set, encompassing 27 samples from various strains and growth conditions sampled at steady state. The absolute protein abundance data were generated based on the QConCAT approach[24,25]. QConCAT employs an isotopically labeled artificial protein containing concatenated peptides of multiple endogenous proteins as external standard to allow for absolute quantification of protein abundance. Using the concatamer to obtain a calibration curve we were able to obtain absolute protein quantification for up to 3708 (median: 3376) proteins (Supplementary Data 1). On average, 28% of the measured proteins were annotated as enzymes and were included in the iCre1355 genome-scale metabolic model (GEM) of *C. reinhardtii*[27] (Fig. 1a). In total, 936 of the 1460 proteins (64%) included as enzymes in iCre1355 were quantified in at least one experimental condition. In comparison, the study employing the largest proteomics data set to date[18], relied on quantification of 840 of the 976 enzymes (86%) annotated in the *S. cerevisiae* model[28]. The large number of enzymes present in *C. reinhardtii* leads to a lower relative coverage, although we quantified the abundance of 11% more enzymes in comparison to what has been attained in *S. cerevisiae*.

We observed a smaller number of quantified proteins in the UVM4 strains compared to CC1690 (Fig. 1a). However, this strain-specific difference is not observable in the total quantified protein amount (Fig. 1b) or mass (Supplementary Fig. 1). While ranking of conditions according to total amount (Fig. 1b) and mass (Supplementary Fig. 1) differ, stress conditions show lower values than mixotrophic standard conditions if only model enzymes are considered (Fig. 1b, Supplementary Fig. 1). Principal component analysis (PCA) of the enzymatic proteins quantified in all samples reveals that replicates cluster together and experiments separate according to strain and culture conditions (Fig. 1c). The first PC resolves strain-specific effects and captures the majority of variance in the data set, while the second PC captures effects specific to the culture condition. Therefore, we concluded that the enzymatic proteins quantified here provide a wide and non-redundant set of *C. reinhardtii's* metabolic states.

### Improved coverage of $k_{app}^{max}$ estimates for *C. reinhardtii*

Our main aim is to make use of the proteomics data to extend the sparse knowledge of enzyme kinetic properties in *C. reinhardtii*. To calculate apparent catalytic rates on a genome scale we used the NIDLE approach that minimizes the number of idle enzymes (i.e., those that do not carry flux, but have abundance measured), representing the principle of effective usage of cellular resources[26]. NIDLE does not rely on maximizing growth as a cellular objective, but rather includes constrains from measured specific growth rates. It is formulated as a mixed-integer linear program (MILP) and does not enforce any proportionality between the measured enzyme abundance and reaction flux. The condition-specific flux distributions obtained by this MILP formulation are then used together with the absolute protein quantification to calculate the apparent catalytic rates, following established approaches[12,26].

Here we expanded on the original NIDLE formulation to calculate estimates of isoenzyme $k_{app}$ values using a linear or quadratic formulation (see Methods). Based on this extension we were able to determine enzyme kinetic data for 18 and 41 reactions with multiple expressed isozymes based on the linear and quadratic formulations, respectively (Supplementary Fig. 2). We decided to use the $k_{app}$ estimates of the quadratic formulation in the following analyses due to the higher coverage. In total, we obtained apparent catalytic rates for 568 enzyme catalyzed reactions (24%) in at least one of the experimental conditions (Fig. 2b, c, Supplementary Data 2), which is the largest set of organism-specific $k_{app}$ estimates generated to date. The previously published pFBA[17] approach together with the QP for isoenzyme $k_{app}$ calculation only resulted in 489 estimates (Supplementary Fig. 3a), that were highly correlated with the NIDLE results (Spearman correlation of log transformed values: 0.96, two-sided *p*-value < 0.0001[29], *n* = 483; Supplementary Fig. 3b). Furthermore, in the NIDLE output for 52% of reactions we were able to calculate $k_{app}$ values in more than half of the investigated conditions. We observed that the largest group (*n* = 189) of $k_{app}^{max}$ values was obtained from all nine considered conditions (Supplementary Fig. 4a), of which most were linked to lipid or heterocycle synthesis (Supplementary Fig. 4b). These results gave us confidence that the maximum over the $k_{app}$ values for a reaction can serve as a good approximation of the in vivo turnover number, since the majority are obtained from all samples that span a range of fluxes. Upon determining $k_{app}^{max}$, we observed the CC1690 and UVM4 standard mixotrophic growth conditions contributed the largest number of reactions operating at the maximum in vivo catalytic rate (Supplementary Fig. 4c). Furthermore, there is no condition that does not contribute information to the calculated $k_{app}^{max}$ values. To provide an overview of the condition- and reaction-wise distribution of $k_{app}$ values, we applied hierarchical clustering to the reaction-wise centered $k_{app}$ values and plotted a heatmap of the values (Fig. 2a). The heatmap indicates that even closely related conditions (according to the distance employed for clustering), such as high cell density and high salt

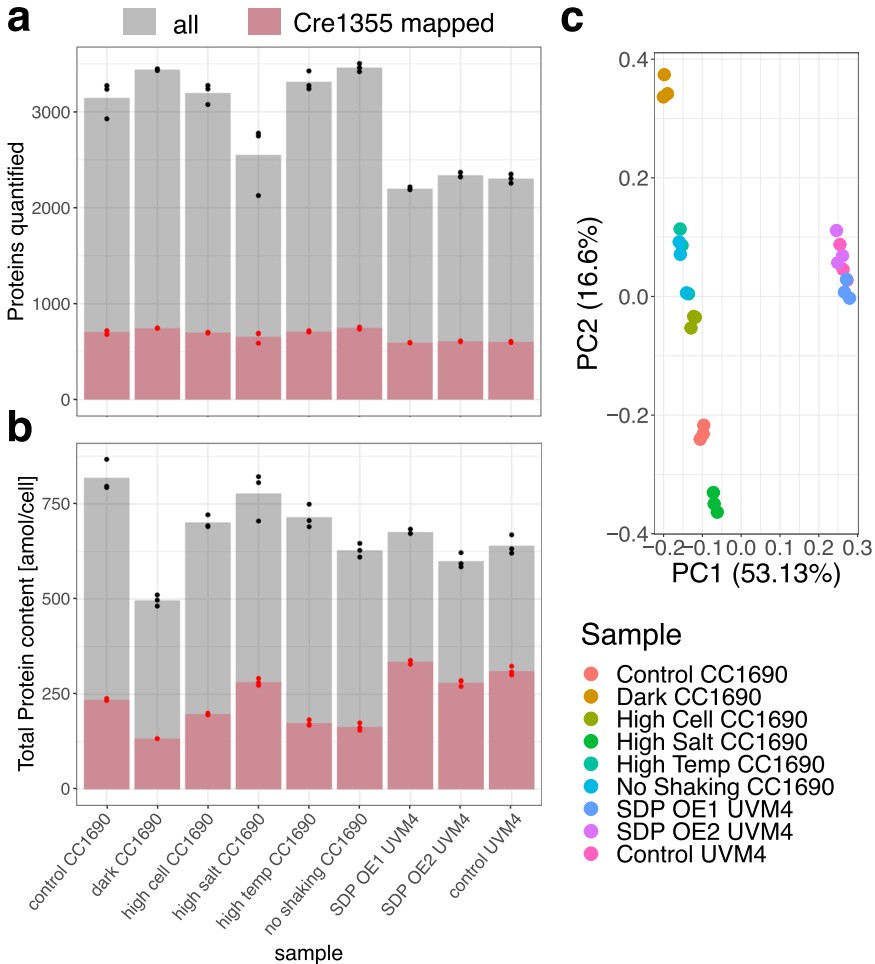

**Fig. 1 | Protein expression and coverage from QConCATdata. a** Number of proteins quantified in at least two of the three replicates per condition, specified in the x-axis. **b** Total protein content summed over all proteins. In panels **a** and **b**, the dark red bar illustrates the subset corresponding to enzymes present in the Cre1355 model. Plotted values are available in the source data. **c** PCA of log-transformed abundance values of enzymatic proteins in *C. reinhardtii*. All replicates in the data set are plotted.

concentration, contain sets of strongly differing $k_{app}$ values (Fig. 2a). Since we did not scale the values, the heatmap also indicates the marked different variance and the row-wise standard deviation (e.g., 75% of the reactions have a standard deviation below $10^2 \, s^{-1}$ but the distribution is skewed with a maximum of $10^{11.2} \, s^{-1}$) (Fig. 2a). This finding indicates that while there are reactions with highly variable catalytic rate, most reactions of primary metabolism show robust $k_{app}$ that only vary by two orders respective to their means. The distinction between samples based on their contribution to $k_{app}$ values is further supported by the PCA using these values (Supplementary Fig. 4d). In contrast to the PCA based on the enzyme proteomic data (Fig. 1c), the largest difference between samples is observed between mixotrophic and heterotrophic growth conditions, resolved by both plotted principal components, while difference between mixotrophic samples is mainly explained by the second principal component (Supplementary Fig. 4d). The PCA of flux vectors obtained from NIDLE can similarly separate samples, though mixotrophic samples show a different separation pattern than in the PCA of $k_{app}$ values (Supplementary Fig. 4e). The differences between the flux solutions are also apparent from the plots of the cumulative sums of the flux distributions (Supplementary Fig. 4f).

The set of $k_{app}^{max}$ values presented here includes reactions from all major subsystems of primary metabolism (Fig. 2c), thus extending the current data on turnover numbers specific for *C. reinhardtii* available at BRENDA[22] and SABIORK[23] about tenfold. Further, for 448 of the

reactions with assigned $k_{app}^{max}$ a query to these databases did not result in any known values in the whole Viridiplantae taxon. When we ranked the metabolic subsystems for which our data provide new enzyme kinetic information, we observed that the largest extension (for Viridiplantae-specific enzymes) was obtained for glycerolipid synthesis and mitochondrial fatty acid elongation (Fig. 2d). Aside from substantially increasing the kinetic information available for this photosynthetic organism, we also provide estimates of in vivo maximum catalytic rate for enzymes that are practically inaccessible to in vitro methods, because they are very difficult to purify and the measurement of reaction rate demands advanced assays. Namely, we were able to determine $k_{app}^{max}$ for 46 transport reactions (top subsystem "Transport, mitochondrial", Fig. 2d) and their respective transporter proteins. Thus, our results provide valuable input for pcGEMs that currently cannot be obtained from existing databases.

### $k_{cat}$ values compiled by GECKO show no correspondence to the estimated $k_{app}^{max}$ values

Studies in *S. cerevisiae*[18] and *A. thaliana*[21] found that in vitro determined turnover numbers provide a rather poor proxy of in vivo turnover number estimates. Thus, we were interested to identify if curated literature $k_{cat}$ values for *C. reinhardtii* correspond to the determined in vivo $k_{app}^{max}$ values. We used the GECKO2.0 heuristic[11,30] to assign the phylogenetically closest available $k_{cat}$ values from BRENDA[22] to reactions (Fig. 2b). For the overlap of reactions that where assigned a

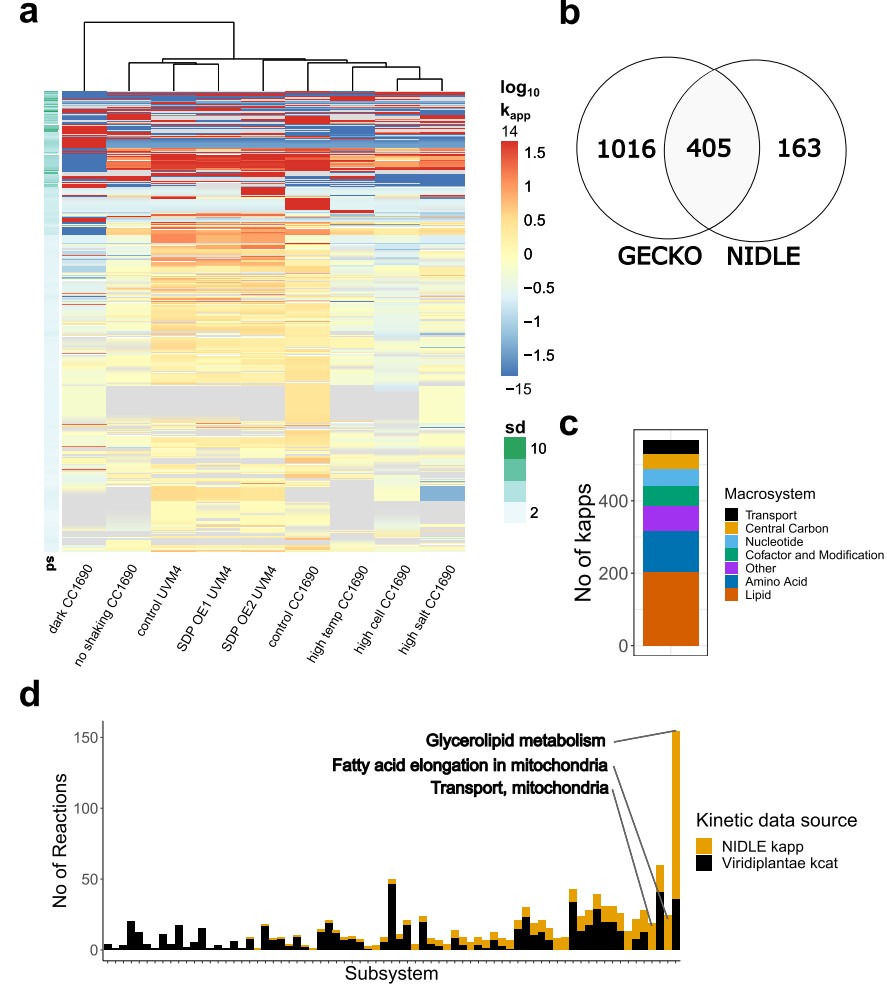

**Fig. 2 | Characteristics of NIDLE-derived estimates of $k_{app}^{max}$. a** Heatmap of $k_{app}$ values for reactions catalyzed by homomeric enzymes or isoenzymes; the values are obtained by applying NIDLE with the physiological and proteomics data of the experimental set-up used. Values were $\log_{10}$-transformed and reaction/row-wise mean centred. Rows and columns were clustered (Methods). Gray fields indicate NA values resulting from conditions where no enzyme quantification was available. The bar on the left indicates the row-wise standard deviation. Prior to log transformation, a pseudo count equal to the minimal value times $10^{-4}$ was added. Only reactions with $k_{app}$ available in at least two conditions were considered ($n = 540$). Values larger than 1.5 and smaller than −1.5 were marked with the most extreme color. Values at the top and bottom of the color legend give the maximum and minimum values rounded to the next higher or lower integer, respectively. **b** Venn diagram showing the overlap in enzyme-catalyzed reactions with maximum $k_{app}$ determined from NIDLE compared and $k_{cat}$ assigned based on EC Numbers by the GECKO heuristic. **c** Stacked barplot indicating the number of $k_{app}^{max}$ values that were determined in the different major metabolic systems of iCre1355[27] GEM of *C. reinhardtii*. **d** The number of reactions with published data on $k_{cat}$ from the Viridiplantae taxon is indicated by a black bar for each metabolic subsystem in iCre1355. The stacked yellow bar indicates the extension of reactions for which $k_{app}^{max}$ value was determined by NIDLE. Raw values are provided in the source data.

maximum catalytic rate by both GECKO and NIDLE, we found that our results corroborate the findings from the two eukaryotes. More specifically, the correspondence between log-transformed values is low (Spearman correlation of 0.19, two-sided *p*-value < 0.001[29], $n = 405$) (Fig. 3a). The scatterplot also reveals that GECKO assigns the same $k_{cat}$ value to many reactions in the lowest quality group, while NIDLE provides specific $k_{app}^{max}$ values that range several orders of magnitude for the same reactions (Fig. 3a). Moreover, the determined in vivo $k_{app}^{max}$ values are systematically lower than the corresponding $k_{cat}$ values (median $\log_{10}$-fold difference (LFD) of −1.4; Supplementary Fig. 5a). Upon subdividing the LFD values according to metabolic subsystems, we found no significant difference between the means of the subsystems (Kruskal Wallace, chi-squared = 7.0, *p*-value > 0.05); however, "Cofactor and Modification" and "Lipid" metabolic systems terms showed larger interquartile ranges then the other terms (Supplementary Fig. 5b). When we investigated the LFD of condition-specific $k_{app}$ to GECKO $k_{cat}$ values, focusing on the subset of reactions for which we obtained a $k_{app}$ in each condition, we found that the heterotrophic

sample have significantly more negative LFDs compared to all the mixotrophic, stress-free conditions (Fig. 3b). This indicates that enzymes in this condition operate at a generally lower rate than indicated by the GECKO assigned in vitro $k_{cat}$ values.

Since the aim of the GECKO approach is to parameterize as many reactions as possible, it iteratively relaxes the matching criteria when assigning $k_{cat}$ values from literature. While we observed ~4-fold higher Spearman correlation with the $k_{cat}$ values for the endogenous *C. reinhardtii* enzymes (0.75, two-sided *p*-value = 0.0019[29] $n = 14$; Supplementary Fig. 5c), the systematic difference between in vivo and in vitro turnover numbers for these enzymes is more pronounced (median LFD: −1.6, Supplementary Fig. 5a). These observations indicate that parameterization with phylogenetic distant enzyme data may be the reason for the low correspondence of ranks of $k_{cat}$ and $k_{app}^{max}$; this, however, does not explain the systematic differences between the compared in vivo and in vitro enzyme catalytic parameters.

It has been shown that eukaryotic pcGEMs parameterized with literature $k_{cat}$ values can yield valuable insights into systemic

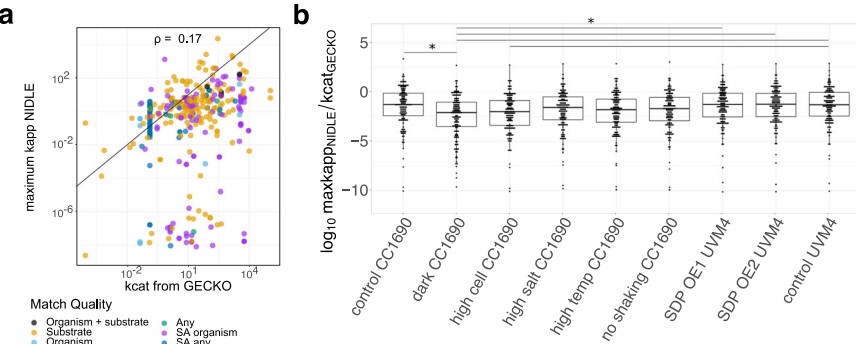

**Fig. 3 | Comparison of in vitro $k_{cat}$ from GECKO with in vivo $k_{app}^{max}$ from NIDLE.**
**a** Scatterplot of the parameter values in the intersection presented in Fig. 2b. The color code gives the matching criteria of $k_{cat}$ values from the GECKO heuristic in order of decreasing stringency. SA specific activity. **b** Boxplot of the $\log_{10}$-fold difference between condition-specific NIDLE $k_{app}$ and GECKO $k_{cat}$ values. Only reactions with a non-zero $k_{app}$ value in all conditions were considered ($n = 144$).

The boxes mark the interquartile range with the bold bar giving the median. Lines give the last data point within 1.5 times the interquartile range. Points in the overlay give the single observations and have been binned. The bars on top indicate pairs of conditions for which the adjusted $p$-value of Tukey's honest significance test was lower than 0.05. Raw values are provided in the source data.

properties of metabolism[31]. Furthermore, for organisms with scarce data on uptake rate and protein abundances, in vitro enzyme parameters remain important resources for developing pcGEM models[32]. However, these results underline once more[18,33] that $k_{app}^{max}$ values are highly preferable parameters if quantitative predictions (e.g., specific growth rate, resource uptake, enzyme usage) are to be generated, since they correct for in vivo/in vitro effects and do not suffer from problematic matching of organism unspecific kinetic data.

## Parameterization of pcGEMs with the estimates of in vivo $k_{app}^{max}$ values show improved enzyme usage prediction

To investigate if the $k_{app}^{max}$ values calculated from NIDLE result in an improvement of the predictive performance of pcGEMs we generated a mixotrophic and a heterotrophic pcGEM for *C. reinhardtii* based on the models published by Imam et al.[27] using the GECKO toolbox[30]. In a first step we used the chemostat data set of Imam et al.[27] to test the effect of the obtained $k_{app}^{max}$ values on growth rate predictions. For each tested condition a so-called raw GECKO model was built, including the $k_{cat}$ values extracted from literature. The over-constraining $k_{cat}$ values were then corrected using the objective control coefficient heuristic and the average enzyme saturation coefficient, $\sigma$, was fitted according to the measured growth rate[27] (Supplementary Table 1). To obtain pcGEMs using the NIDLE $k_{app}^{max}$, the $k_{cat}$ values in both the raw and the corrected GECKO models were substituted with the respective $k_{app}^{max}$ estimates, were available.

Although the recent GECKO2.0 study[30] shows that raw models do not produce biological meaningful predictions, we included them in the performance assessment since we: (i) deemed the comparison interesting for scenarios in which pcGEMs of photosynthetic eukaryotes without available chemostat growth data are built, preventing the application of the GECKO correction procedure unavailable, and (ii) aimed to disentangle the effect of GECKO correction from the effect of $k_{app}^{max}$ parameterization. In line with the reports of raw model performance in heterotrophic organisms[30], we found that the raw GECKO models of *C. reinhardtii* with and without usage of $k_{app}^{max}$ underestimate growth compared to FBA predictions (Fig. 4a). The only exception was the heterotrophic conditions, in which NIDLE raw pcGEM predicts higher growth then experimentally observed. In all cases, the experimental growth rate was reached only after the $k_{cat}$ correction step and refitting $\sigma$. For heterotrophic conditions, $\sigma$ was fitted to ~0.4 of the value in autotrophic in mixotrophic conditions (Supplementary Table 1), indicating that in heterotrophic growth many enzymes are expressed considerably higher than necessary to maintain metabolic flux (Fig. 4a). This finding recapitulates the previous results of significantly larger negative differences to $k_{cat}$ values in the

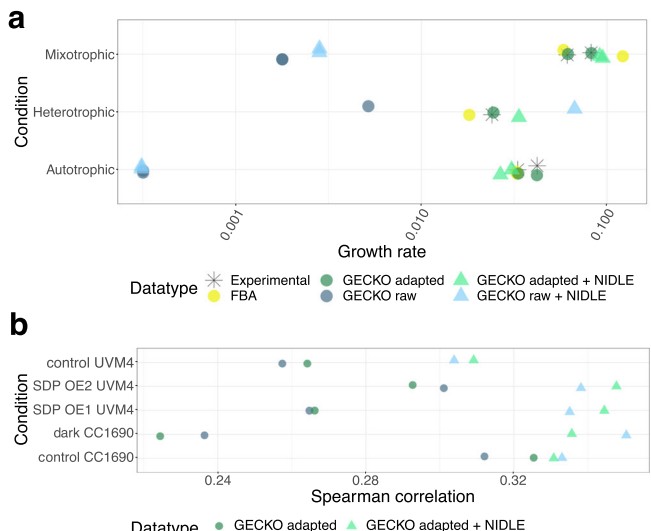

**Fig. 4 | Prediction performance of enzyme usage with different pcGEM model parameterizations. a** Comparison of experimental data from chemostat cultures[27] and predictions from FBA and pcGEMs parameterized with uncorrected $k_{cat}$ values obtained from BRENDA[22] (GECKO raw), corrected $k_{cat}$ values using GECKO heuristic (GECKO adapted) or updated with enzyme wise $k_{app}^{max}$ from NIDLE (+NIDLE). Mixotrophic and autotrophic growth was measured in duplicates; heterotrophic growth is a single measurement. **b** Spearman correlation of predicted enzyme usage based on pcGEMs and observed enzyme abundance in QConCat data set in $\log_{10}$-scale. The tested condition was not considered when calculating the $k_{app}^{max}$ values from NIDLE. Plotted values are provided in the source data.

independent heterotrophic growth experiment of the proteomics data set (Fig. 3b). Comparing the performance of NIDLE pcGEMs we did not observe a strong effect on growth rate predictions. As expected, the prediction error was higher than in the corrected GECKO pcGEMs, since the latter were fitted to the experimental growth rate; however, the introduced error (RMSE 0.0163) was comparable to that of canonical FBA (RMSE 0.0183) (Fig. 4a).

Since the maximum catalytic capacity used in pcGEMs quantifies the enzymatic expenditure to support a certain reaction flux, another important application of these models is in predicting the allocation of total enzyme mass into specific enzymes. Therefore, we tested the effect of the $k_{app}^{max}$ values obtained by NIDLE on the accuracy of enzyme usage prediction in the standard conditions included in our proteomics data set. To allow for an informative comparison, we left out

the NIDLE $k_{app}$ values calculated in the tested condition when calculating $k_{app}^{max}$ for the respective NIDLE pcGEMs (see Methods). We predicted enzyme usages by fixing the flux through biomass reaction to the experimentally observed growth rate and minimizing the total enzyme mass. Next, we calculated the Spearman correlation (Fig. 3b) and Root Mean Squared Error (RMSE) (Supplementary Fig. 6) between the predicted and the measured enzyme abundances. Interestingly, we observed that models parameterized with $k_{app}^{max}$ from independent proteomics samples showed higher predictive performance in the unseen condition than the canonical GECKO models. The Spearman correlation improved for mixotrophic and heterotrophic test conditions to ~0.34 (except for control UVM4, which showed a slightly lower improvement) (Fig. 3b). The RMSE was reduced by ~30% upon the integration of $k_{app}^{max}$ values, although in this scenario the heterotrophic test condition showed systematic higher errors than mixotrophic test scenarios. This observations held irrespective of whether the raw or corrected GECKO model was compared (we note that the corrected GECKO model is fitted to the experimental growth rate from mixotrophic chemostat experiments). Thus, our results demonstrated that the NIDLE approach successfully uses information from physiological data to calculate maximum catalytic capacity values which are a better predictor of enzyme usage than widely used literature values.

## Matched proteomics and metabolic flux analysis will improve in vivo $k_{cat}$ estimation

We presented a protein abundance data set with extensive coverage of the proteome response to various perturbations. This data set comprises a valuable resource for systems biology studies in *C. reinhardtii*. Here we made use of this resource to considerably expand the information for green algae. Due to the extended NIDLE formulation we were able to estimate 568 (24% of enzymatic reactions) $k_{app}^{max}$ values for enzyme catalyzed reactions, compared to 378 (14%) and 358 (10%) previously reported $k_{app}^{max}$ values in *E. coli*[17] and *S. cerevisiae*[18], respectively. To our knowledge, we present here the data set with the highest $k_{app}^{max}$ coverage available to date. These previous studies used metadata sets that assembled protein quantifications across different studies whereas our entire data set was measured on the same setup using the QconCAT approach to obtain absolute protein abundances. While this reduces systematic error between different experiments in the data set it limits the number of sampled conditions due to available resources. We aim to extend the presented QconCAT data set with further complementary measurements in the future. Nevertheless, the predictions in unseen test conditions (Fig. 4b) indicated that we are able to obtain robust improvement of model accuracy with the estimates obtained from the nine different conditions used in this study. The obtained kinetic parameters allow to quantify the costs for different cellular pathways and thus foster the application of advanced metabolic engineering strategies[13] in the biotechnologically relevant taxon of green algae. The next step to further improve the accuracy of estimated parameters is to integrate metabolic flux measurements from isotopic labeling of the investigated culture conditions. This methodical milestone has not yet been achieved in eukaryotes, where in vivo $k_{cat}$ estimation studies relied solely on constrained based modeling[17,18,21]. However, the feasibility of such a study has been proven in *E. coli*[19]. Recent methodical improvements in *C. reinhardtii* metabolic flux analysis[34] put this promising combination of experimental data in reach. Future interdisciplinary efforts to improve our understanding of photosynthetic metabolism in green algae and their descendants will focus on achieving this milestone.

## Methods

### Data set assembly

Analyzed *C. reinhardtii* data sets included data from previously published QCONCAT studies available under PRIDE[35] data set identifier PXD018833 (Control UVM4, SDP OE1 UVM4, SDP OE2 UVM4) [https://www.ebi.ac.uk/pride/archive/projects/PXD018833] and were further augmented by data sets measured as part of this study and made publicly available under the PRIDE identifier PXD037599 (Control CC1690, Dark CC1690, High Cell CC1690, High Salt CC1690, High Temp CC1690, No Shaking CC1690)[https://www.ebi.ac.uk/pride/archive/projects/PXD037599]. As control cultivation conditions for *C. reinhardtii* CC1690 cells, cultures were grown for 48 h in tris-acetate-phosphate (TAP) medium using a rotatory shaker operating at 2 turns per second, while being constantly illuminated at 100 μmol photons m$^{-2}$ s$^{-1}$ at and held at 24 °C. Data sets differing from control conditions were created by alteration of listed growth parameters (a detailed description of modified factors is available in Supplementary Data 3).

### QconCAT expression and protein purification

The coding sequence for the PS-Qprot protein was codon-optimized for *E. coli* expression and synthesized by Biocat (Heidelberg), incorporating BamHI/HindIII restriction sites for subsequent cloning steps. The synthesized PS-Qprot coding sequence was cloned into the pET-21b expression vector (Novagen) using the BamHI and HindIII restriction sites. M9 minimal medium was prepared with 9.2 mM 15NH4Cl (98%, Cambridge Isotope Laboratories) and 100 μg ml−1 ampicillin. The transformed E. coli ER2566 cells harboring pET-21b-PS-Qprot were inoculated into the 15N-labeled M9 minimal medium. The culture was incubated under appropriate conditions to allow protein expression and 15 N incorporation. After sonication, supernatant was applied to a Co-NTA column (G-Biosciences) for purification based on the His-tag present on the PS-Qprot protein. The column was washed three times with Urea Buffer (8 M urea, 20 mM Tris-HCl pH 8, 0.25 M NaCl) containing 5 mM, 25 mM, and 100 mM imidazole, respectively. The eluted protein was concentrated and dialyzed into phosphate buffered saline. The concentration of the PS-Qprot protein was determined spectroscopically at 280 nm using a NanoDrop spectrophotometer. The concentration was calculated based on the Lambert-Beer's law, assuming a molecular weight of 39,945.63 g/mol for PS-Qprot and an extinction coefficient of 86,860 M$^{-1}$ cm$^{-1}$[25]. The extinction coefficient values were determined using the ExPASy ProtParam tool.

### LC−MS/MS measurement and raw data analysis

After cell harvesting and protein extraction, all samples were spiked with a master mix of Chlamydomonas-specific QConCAT proteins and digested tryptically. Peptides were separated using a reversed phase chromatography system consisting of a trapping column (Triart C18, 5 μm particles, 0.5 mm × 5 mm) and an analytical column (Triart C18, 3 μm particles, 300 μm × 150 mm, YMC). The separation was performed at a flow rate of 4 μl/min. The chromatographic separation was achieved by employing a gradient elution method with HPLC buffer A (2% acetonitrile, 0.1% formic acid) and HPLC buffer B (90% acetonitrile, 0.1% formic acid). The gradient profile ranged from 2% to 35% of buffer B over the course of the separation. Measurement was performed via LC−MS/MS (Eksigent nanoLC 425 coupled to a TripleTOF 6600, ABSciex)[25]. Quantitative analysis of MS/MS measurements was performed using ProteomIQon 0.0.7[36]. Peptide searches were performed upon the assembly of a peptide database based of the *Chlamydomonas* proteome based on version JGI5.5 of the *C. reinhardtii* genome blended with the sequences of spiked-in QconCAT proteins. The search space included methionine oxidation and acetylation of protein N termini as variable modifications and was extended by $^{15}$N-labeled variants of *Chlamydomonas* proteins. False discovery rate thresholds for peptide spectrum matches and protein group identifications were set to 1%. Following peptide spectrum matching, ion abundances were estimated by integration of the areas of extracted-ion chromatograms.

### QConCAT-based estimation of absolute protein abundances

To obtain absolute protein abundances, we first aggregated ion species to the modified peptide level by summation (e.g., different charge

states). Differently modified versions of the peptides were aggregated to the peptide and then protein-group level by median-based aggregation, yielding preliminary protein abundance estimates. Computing the ratio between native, unlabeled peptides and $^{15}$N-labeled peptides originating from spiked-in QConCAT proteins allowed to estimate absolute protein abundances for a multitude of different *C. reinhardtii* proteins[25], as previously described[37,38]. With these high-quality Qcon-CAT-based abundance measurements at hand, we were able to create calibration curves by regressing the latter on the preliminary protein abundance estimators, and thus to compute proteome-wide absolute abundance estimates.

## Processing of non-proteotypic peptides

A data set entry is considered to have ambiguous entries if its quantification is based on a peptide that is non proteotypic (i.e., is present in multiple proteins); otherwise, the protein is defined to have an unambiguous entry. For ambiguous entries, an iterative approach was used to remove them in each sample. If an unambiguous entry of one of the mapped proteins was present, the corresponding concentration was subtracted from all ambiguous entries of this protein and the protein ID was removed from the ambiguous entries. If the cellular concentration of an ambiguous entry was smaller than 0 after subtracting, the entry was removed from the sample. This procedure was repeated until no further protein IDs could be removed from ambiguous entries. The remaining ambiguous entries were removed from the sample data. Proteins that were only quantified in one of the three biological replicates were removed from the data set. For the remaining data the median over the measured replicates was used in the downstream analyses.

## GEM used in constraint-based modeling

The most recent SBML and COBRA compatible model of *Chlamydomonas reinhardtii* "iCre1355"[27] was employed. The erroneous reaction formular of 'CAT' was updated to "2 h2o2[c] → 2 h2o[c] + o2[c]". GPR rule syntax was updated to not include "... and (GENE1 or GENE2 ...) ..." rules. All model modifications and mathematical programs solved in this study was carried out using the COBRA toolbox[39] and GUROBI solver[40] in MATLAB[41].

## NIDLE

We used the iCre1355 mixotrophic and heterotrophic model[27] with the respective culture conditions used in the proteomics experiments. The NIDLE approach is formulated for positive, real valued flux variables; therefore, the models were converted to irreversible by splitting each reversible reaction into irreversible forward and backward reactions. All uptake reactions were constrained by the model supplied bounds except for acetate uptake. For mixotrophic conditions a linear regression model was fitted based on the mixotrophic chemostat culture measurements from Imam et al.[27], in which acetate uptake rate was predicted based on growth rate. The model was fitted using the R *lm* function[42] with default options, and the model predicted uptake rate increased by the standard error of prediction was used as an upper bound on the acetate uptake rate for the mixotrophic culture scenarios[27]. For the heterotrophic condition the maximum measured acetate uptake rate from the Imam et al. heterotrophic chemostat data was set as an upper bound (Supplementary Table 2). We adapted the source code in the NIDLE repository[26] to the iCre1355 model, but the formulation of the NIDLE approach, based on a MILP, remained unchanged.

To calculate $k_{app}$ values for homomeric isoenzyme catalyzed reactions we first determined if only one of the catalyzing isoenzymes is quantified in the respective condition. If this was the case the $k_{app}$ was calculated in the same way as for the homomeric enzymes, i.e., the reaction flux of reaction $i$ in condition $j$ divided by the respective

enzyme abundance $E$ gives the apparent catalytic rate,

$$k_{app_{i,j}} = \frac{\mathbf{v}_{i,j}}{\mathbf{E}_{i,j}} \tag{1}$$

To convert enzyme abundances measured in amol/cell to mmol/gDW the literature cell dry weight of 48,000 pg[43] was used for a mixotrophic grown cell and the dry weight of other conditions was calculated from measured cell volume assuming constant dry weight density.

In the case that multiple isoenzymes have been measured by mass spectrometry we integrated information from different conditions to decompose the contribution of different isoenzymes to the observed flux. We took advantage of the fact that in the mixotrophic standard growth conditions best approximate the maximum apparent catalytic rate for the majority of enzymes (Supplementary Fig. 4d), and assumed equal $k_{app}$ values for an isoenzyme in the different conditions. This allowed us to formulate a quadratic problem based on the flux predictions and enzyme abundance measurements in the four mixotrophic standard conditions (i.e., Control CC1690, Control UVM4, SDPOE1 UVM4, SDPOE1 UVM4, SDPOE2 UVM4), given in the following

$$\min \sum_{j \in C_{std}} \boldsymbol{\delta}_j^2 \, s.t. \tag{2}$$

$$\sum_i \left( \mathbf{E}_{i,j} * \mathbf{k}_{app_i} \right) + \boldsymbol{\delta}_j = \mathbf{v}_j \tag{3}$$

$$k_{app} \geq \varepsilon \tag{4}$$

More specifically, we obtain $k_{app}$ estimates by minimizing the quadratic sum of residuals between flux supported by the $k_{app}$ values and obtained from NIDLE, over all conditions j. We chose $\varepsilon$ of $10^{-10} \cdot 3600 \, [\text{h}^{-1}]$ since both the smallest turnover number in the joint public databases $(5.8 * 10^{-10} [\text{s}^{-1}])$[22,23] and calculated from homomeric reactions $(4.0 * 10^{-10} [\text{s}^{-1}])$ were in this order of magnitude. The effective reaction-specific $k_{app}$ for each condition was then calculated as the average weighted by the protein abundance in the given condition,

$$\mathbf{k}_{app_j} = \frac{\sum_i \left( \mathbf{E}_{i,j} * \mathbf{k}_{app_i} \right)}{\sum_i \mathbf{E}_{i,j}} \tag{5}$$

We also compared the solution from minimizing the $\ell_1$-norm of the error term, $\boldsymbol{\delta}$,

$$\min ||\boldsymbol{\delta}||_1 \tag{6}$$

subjected to the same constrains (Supplementary Fig. 2a). We did not consider $k_{app}$ values equal to the lower bound, $\epsilon$.

## pcGEM creation using the GECKO toolbox

The GECKO toolbox[11,30] was used to integrate maximum catalytic rate data into a pcGEM. Based on each of the published models (mixo-auto-, and heteroptrophic), and the obtained chemostat experiments[27] corresponding pcGEMs were created. Scripts were adapted according to the README instructions (https://github.com/SysBioChalmers/GECKO). For compatibility with the GECKO toolbox, the JGI gene ids in iCre1355[27] were converted to Uniprot ids and introduced duplicates where removed. The protein content used for biomass rescaling and limiting of the enzyme pool reaction was taken from the measurements of Boyle & Morgan[44]. For all pcGEM simulations the uptake rate bounds of the macronutrients ammonium, phosphate, and carbon dioxide where set to $1000 \frac{\text{mmol}}{\text{gDW} \bullet \text{h}}$. The average protein abundance over all sampled conditions was used to calculate the factor $f$ (only proteins without missing values were used). Growth associated maintenance was not refitted. A corrected model based on the observed chemostat growth measurements in the model publication[27] was created using GECKOs objective control coefficient heuristic to correct over

constraining $k_{cat}$ values, and the sigma factor was fitted. The NIDLE pcGEMs were generated by substituting GECKO-assigned $k_{cat}$ values for each enzyme pseudometabolite in the augmented stochiometric matrix with the maximum of all NIDLE obtained $k_{app}^{max}$ calculate over all reactions this enzyme catalyzes (Supplementary Data 4). For the comparison of growth rate predictions the respective biomass reaction was used as objective function.

The pcGEM fitted with chemostat data from "Mixotrophic_Rep3" and "Heterotrophic_Rep1" were used for the simulation of enzyme usage in the proteomics experiments of the respective growth scheme. In the NIDLE models used for enzyme usage comparison the substituted $k_{app}^{max}$ values were calculated omitting the $k_{app}$ values obtained from the tested condition. The same condition specific uptake flux constraints as in the NIDLE problems were used. Flux trough the biomass reaction was fixed to 0.99 of the observed growth rates and the flux through the "draw enzyme pool" reaction was minimized.

### Querying the BRENDA and SABIO-RK database
Turnover numbers of non-mutated enzymes together with organism and EC-number information were downloaded as text files from BRENDA[22] and SABIO-RK[23] databases (status 07/2022) and joint. For the reactions with EC-number annotation in iCre1355[27] the following matching criteria for enzymes with fully matching EC-number where tried in the given order:

1. Chlamydomonas taxon & substrate
2. Chlamydomonas taxon
3. Viridiplantae taxon & substrate
4. Viridiplantae taxon

The maximum of all $k_{cat}$ values in the first criteria with non-zero number of matches was assigned as comparison $k_{cat}$.

### Clustering of $k_{app}$ values
A pseudocount of $10^{-4}$ times the lowest estimated $k_{app}$ value was added to all values prior to $\log_{10}$-transformation. $k_{app}$ values were reaction-wise mean centered. The Euclidian distance between the $k_{app}$ vectors was calculated using the R function dist() from the stat package[42]. In case two compared vectors didn't have a single pair of comparable entries due to NA values resulting from missing enzyme quantifications, the distance was set to $10^{50}$. Hierarchical agglomerative clustering using the single linkage approach was applied as implemented in the R stat package[42].

### Statistics and reproducibility
Details on the statistical approaches used are given in the figure captions and methods section. Proteomics data were acquired in triplicates to assess reproducibility. For all other statistical tests sample size is given in the text or figure caption. No statistical method was used to predetermine sample size. Peptides that were only determined in one replicate were excluded from the analysis. The experiments were not randomized. The Investigators were not blinded to allocation during experiments and outcome assessment.

### Reporting summary
Further information on research design is available in the Nature Portfolio Reporting Summary linked to this article.

### Data availability
The proteomics data used in this study have been deposited in the PRIDE database[35] under accession codes PXD018833 (UVM4 data set) [https://www.ebi.ac.uk/pride/archive/projects/PXD018833] and PXD037599 (CC1690 data set) [https://www.ebi.ac.uk/pride/archive/projects/PXD037599]. The $k_{app}$ estimates generated in this study are provided in the supplementary files (Supplementary Data 2 and 4). Source data are provided with this paper.

### Code availability
Code for the updated NIDLE approach and generation of the presented results is publicly available as GitHub repository[45].

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

## Acknowledgements

Z.N. would like to thank the Research Focus Group "Evolutionary Systems Biology" of University of Potsdam for funding. Z.N., and M.A. would like to thank the Max Planck Society for funding. Z.R. was supported by the European Union's Horizon 2020 research and innovation program grant 862201 (to Z.N.) (this publication reflects only the author's view and the Commission is not responsible for any use that may be made of the information it contains).

## Author contributions

M.A. and D.Z. performed research. D.Z. and F.S. run experiments and acquired protein abundance data. M.A. wrote code and analyzed data, R.X. provided code and troubleshooting support for NIDLE. M.A., D.Z., T.M., and Z.N. designed research. M.A., Z.N., D.Z., and T.M. wrote the paper.

## Funding

## Competing interests

The authors declare no competing interests.
