## [Peer Review File · Nature Communications]

Proteomics and constraint-based modelling reveal enzyme kinetic properties of *Chlamydomonas reinhardtii* on a genome scaleREVIEWER COMMENTS

Reviewer #1 (Remarks to the Author):

Arend et al. Present a global study of kinetic parameters in the metabolism of the green algae *Chlamydomonas reinhardtii*. There is much interest in getting new insight into algae as they can contribute to greenhouse gas mitigation in terms of converting carbon dioxide to food, fuels and chemicals. In this study a comprehensive dataset of semi-absolute protein concentrations across 9 conditions, obtained by merging previous published data with additional new measurements, is made available. The culture conditions span autotrophic, mixotrophic and heterotrophic metabolism. This study uses a genome-scale metabolic model and physiology data to compute metabolic fluxes using a MILP approach (NIDDLE). Next, maximum apparent enzyme turnover numbers (k_{app}^{max} , which reflect the maximum capacity of an enzyme at in vivo conditions) are calculated assuming proportionality between the computed fluxes and the measured protein abundances. The authors highlight their large contribution to the extension of data on kinetic parameters for green algae's metabolism (from 85 parameters to 568 entries). The computed kinetic parameters provide a high number of entries in lipid metabolism and mitochondrial transport, which were not covered by the values in databases.

k_{app}^{max} values are compared to the enzyme turnover numbers matched by the GECKO method to each of the reactions of a protein-constrained version of the GEM, finding that GECKO values are systematically higher than the computed k_{app}^{max} , while the latter offer a catalogue of condition dependent catalytic rates. Their results support previous findings reporting that in vitro kinetic parameters do not match k_{app}^{max} values computed from metabolic fluxes and proteomics in eukaryote organisms. Different protein-constrained models are generated, based on the GECKO formalism, using kinetic parameters from different origins (computed k_{app}^{max} and literature k_{cat} assigned by GECKO). Such models are used to evaluate predictive performance on maximum growth rate and protein usage across several conditions. The authors conclude that the use of computed k_{app}^{max} for parameterization of protein-constrained models improves quality of phenotype predictions.

Evaluation

The manuscript is generally well written and concise. Overall, the study provides valuable results that contribute to extend the available knowledge of *Chlamydomonas reinhardtii* metabolism, using a computational approach, aided by generation of high-throughput data. One could have wished for a more in-depth analysis of the physiology of the organism using the model, i.e. novel biological insights, but in light of the large amount of work carried out this is not a key issue. Therefore, I consider that the quality and relevance of this work is suitable for publication in Nature Communications after the following major and minor points are addressed.

Major points:

- 1.- Lines 163-167: Here the details for the comparison between GECKO k_{cat} s and computed k_{app}^{max} is detailed. Low correspondence of values is explained here just by a low spearman correlation coefficient, which is a measure of how monotonic the relation between two distributions of values is. Generalized and Michaelis-Menten enzyme kinetic expressions clearly show that the capacity limit of enzymes (k_{cat}^{max}) is only one of several terms/factors affecting reaction rates, from this it is expected that in vivo capacities differ from those in vitro to an extent that is dictated by context dependent properties. This is in line with the sentence indicating that GECKO k_{cat} values are systematically higher than computed k_{app}^{max} . But to what extent? It is suggested to characterize the discrepancy

between these two distributions in quantitative detail. For instance, statistics on ratios between log₁₀ values from the two distributions will inform of how many orders of magnitude are GECKO values higher than k_{app}^{max} , a relevant issue when thinking of kinetic parameters as constraints for metabolic flux predictions. Additionally, how large are those differences in enzymes grouped by metabolic context (subsystems)? With such an analysis, focusing on *C. reinhardtii* or taxon-specific values, other biological features can be explored, which may be of interest to an audience even broader than metabolic modelers, for instance, for instance pathway specialization of enzymatic regulation and control in *C. reinhardtii*.

2.- Lines 173-174: Here the authors state that literature turnover numbers are a suboptimal source of parameters for pcGEMs. This is an issue that is hard to generalize, as pcGEMs may serve for different purposes, as any other GEM they are a comprehensive knowledgebase of an organism's available information. For some applications predictive performance is important, in such cases the use of k_{app} is relevant for refining predictions (explored in the next section). However, literature turnover numbers are still an important source of parameters for non-model organisms for which exchange fluxes and proteomics data are scarcely available. Heuristic kinetic parameterization of pcGEMs has been used for quick generation of models for poorly characterized organisms and cell types, and have been helpful at providing general insights of their metabolism (Robinson et al. 2020; Lu et al. 2021).

3.- In the section "Parameterization of pcGEMs with the estimates of in vivo k_{app}^{max} values show improved enzyme usage prediction" performance comparisons between three models are performed. pcGEM with NIDLE parameters, a raw GECKO model and a calibrated GECKO model. Nevertheless, in the most recent publication of GECKO it is stated that the use of the "raw model" for quantitative simulation should be avoided, as this kind of model serves as a scaffold for proteomics data integration, therefore, only the calibrated GECKO model should be compared to the one with NIDLE-origin parameters. The sentence in lines 188-189, formulated as a finding, highlights what has been said previously and was the main motivation for the introduction of a parameter calibration procedure into GECKO2.0.

4.- Line 192: It was found that the fitted sigma factors differ across conditions, in GECKO sigma represents a lumped or average effect of condition-dependent factors over enzyme capacity. In contrast k_{app}^{max} values offer the opportunity of estimating such effects for each particular enzyme for which a *C. reinhardtii*, or related, K_{cat} value exists. In order to provide a more detailed analysis of this, please compare statistical metrics (e.g. median, mean, coefficient of variance) for the distribution of ratios between k_{app}^{max} and K_{cats} across conditions versus the different sigma values.

5.- Lines 194-197: Growth rates are usually used as a calibration parameter for pcGEMs in GECKO, therefore, it is not directly a prediction from which predictive performance can be evaluated and compared to other approaches. It is suggested to use another prediction output (not used in calibration of the models) for comparison with the NIDLE parameters-based model.

6.- Lines 215-217 and Figure 3B: Predictive performance of models is compared in terms of spearman correlation between simulated enzyme usages and measured protein abundances. This is a metric of correlation, however, it does not inform about how accurate predictions are. Please include other error metrics in analysis.

Additionally, computation of k_{app}^{max} values depend on input data. It is said that the tested condition was omitted from the calibration data for each case, however, if the training datasets are very similar to each other, then it is expected that predictions are good even when omitting the tested conditions. In figure 1 a PCA shows the differences at the protein

expression level between conditions, additionally, an analysis of the different conditions in terms of both, used flux constraints, and simulated flux distributions is also required, in order to assess how different are the calibration data from the outputs of the tested condition. This will inform if observed predictive performance is somehow artificial or not.

7.- It is recommended to offer more information about the condition dependency of computed k_{app}^{max} values, which is one of the major contributions of this paper. What is the variance and statistics of k_{app}^{max} across conditions, focusing on single enzymes. Obtention of genome-scale trends of condition dependency of catalytic rates for Eukaryotes might be a valuable resource for systems understanding of gene and protein regulation and control mechanisms.

8.- k_{cat}^{\wedge} values offer an estimate of the physico-chemical limits of enzymes, for some applications it is important to know what are the limits for fine-tuning enzymes' performance, in this contexts k_{cats} become relevant. However, the general tone of the manuscript leaves the impression of dismissing k_{cats} measurement, it is suggested to review the article and make sure that a nuanced discussion about pros and cons of both the use of k_{cat}^{\wedge} and k_{app}^{max} . For what kind of applications one may be more useful and/or informative than the other in the context of pcGEM?

Minor points:

8.- Line 36: "However, economic feasibility of photosynthetic bioreactors requires further optimization of desired biotechnological objectives", provide examples of the typical uses of "biotechnological objectives" that this algae has been used for, it will provide context for understanding the relevance of this study to non-modelers.

9.- Line 44: New successful studies of metabolic engineering targets identification with protein constrained models are now available and published (e.g. targets for different products using a *B. subtilis* pcGEM, DOI: 10.3390/microorganisms11010178, and heme using a *S. cerevisiae* model, DOI: 10.1073/pnas.2108245119). The introduction focuses mostly on pcGEMs and kinetic parameters but a bit more of context is needed, so that a general audience can comprehend the relevance of this work and the extension of the field.

10.- Line 48: A single study (also by the authors) is cited as the only reference regarding computation of apparent enzyme catalytic rates. As the topic is a current active discussion with plenty of development in recent years, please add additional references, these parameters have previously been computed using flux balance analysis, resource balance analysis and ME-models for different organisms (*E. coli*, *S. cerevisiae* and *B. subtilis*). This will clarify the story of development in the field to the reader, especially to those not completely focused on FBA approaches, while also offering supporting evidence for the conclusions of this manuscript.

11.- Line 56-58: It is said that in order to make pcGEMs for green algae, a substantial increase of in vivo turnover numbers is needed. However, the current available tools (sMOMENT, ECMpy, GECKO) already enable integration of this kind of constraints to any standard metabolic network, rather, apparent in vivo kinetic parameters are needed for refining predictions of such models. It is suggested to edit the sentence accordingly.

12.- Line 70: The term "unseen test conditions" is confusing in this context, as it requires further explanation to be understood, which is done below in the text. I suggest removing this from the sentence as this is an introduction, and the sentence already says that predictive accuracy of enzyme resource allocation was improved.

13.- Line 73: Correct sentence to : "we employed a comprehensive, ..."

14.- Line 80: The first time that the model iCre1355 is mentioned, however, the article referring to its original development is not cited here, which may confuse some readers and make them think that iCre1355 construction is also part of this manuscript's work. Please cite.

15.- Lines 82-84: The absolute number of enzymatic proteins between two different studies, that used different quantification methods and based on different organisms, is directly compared here, highlighting that the current study measured 100 more proteins than the previous. As proteomes differ drastically in size and composition, this kind of comparison does not reflect the "depth" of the studies. Rather, it is suggested to compare the extent of measured enzymatic proteins between different studies based on the percentage of measured enzymes out of the total number of known enzymes for a given organism (coverage), or enzymes in their models in this case.

16.- Figure 1 captions: The order of the subfigures in the description (a, b, c) does not correspond to the order in the figure.

17: Figure 1b: The total protein measured per sample is quantified in term of amol/cell. Then, these quantities are compared across conditions, showing notorious differences with no systematic patterns. What if in a given stress condition a cell needs to highly overexpress a few stress-coping proteins, which have been found to tend to be relatively light proteins (Doughty et al. 2020), while keeping its central carbon metabolism enzymes (which tend to be heavier and less likely to be significantly differentially expressed) mostly unchanged? This situation may lead to an scenario that shows increased number of total protein molecules, but may not reflect the fact that such changes are not significant in a genome-scale. It is suggested to change representation of this figure by comparing protein mass (pg/cell) across conditions and reevaluate the argument that no systematic patterns of total protein expression were observed across conditions (written in lines 94-95).

18.- Line 117: "which is the largest set of organism-specific k_{app} estimates generated to date". The statement highlights technical relevance, but not necessarily biological relevance, it is suggested to normalize the number of quantified proteins and or protein enzymes, according to the number of known proteins in *C. reinhardtii* and/or number of known enzymes. It may also be convenient to add a comparison on the number of studied conditions and strains, which may emphasize the scale of this study.

19.- Line 120: Specify the statistical test used for computing p-value and if it is one or two-tailed.

20.- Line 121-123: Additional analysis of the core computed k_{app} (group of $n=189$ obtained for the 9 conditions) I recommended, for instance in which sectors of metabolism or pathways do they participate? This kind of analysis will inform about the swaths of cell metabolism that are being explored and shed light on by this study, which is said to be a significant increase of knowledge for *C. reinhardtii*.

21.- Lines 123-124: It is mentioned that results, until this section, suggest that selection of a maximum k_{app} across conditions, k_{app}^{max} , serve as a good approximation for in vivo turnover numbers. What is it meant by this? It is suggested to express this in terms of performance metrics. Additionally, so far in the manuscript, k_{app}^{max} values have been compared between two different computation schemes (NIDLE and pFBA) this may inform about prediction precision, but not accuracy, and also in terms of coverage, however this is not informative of how good the approach is in terms of reflecting in vivo conditions.

22.- Line 126: "operating at the maximum catalytic rate...", as this the same term that it is

used to describe k_{cat} , it is suggested to be substituted by “maximum in vivo catalytic rate”.

23.- Lines 138-139: Check writing in point C in figure 2 captions. Not clear.

24.- Figure 2D captions: please indicate what $\rho = 0.17$ means here.

25.- Line 142: It is suggested to change “of decreasing quality” to “of decreasing stringency/specificity”, as evaluation of a match quality requires comparison to an in vivo known value, case by case. However, the phrasing could be kept if different correlation values, or error metrics between GECKO k_{cats} and computed k_{app}^{max} , are estimated for the different groups of parameters matched by GECKO.

26.- Line 152: same suggestion as point number 22.

27.- Lines 159-160: Change “in vitro determined turnover numbers provide a rather poor proxy of in vivo turnover numbers” to “in vitro determined turnover numbers provide a rather poor proxy of in vivo turnover numbers computed from protein abundance and estimated fluxes” or something similar. Since in vivo turnover numbers cannot be directly measured, therefore, their exact value is not available.

28.- Lines 212-213: Avoid using the term “enzyme usage coefficients”, as the word coefficient is usually used for non-variable values. Instead, enzyme usages are state variables of protein constrained models.

Reviewer #2 (Remarks to the Author):

In the manuscript by Arend et al., the authors employed a protein-constrained genome scale metabolic model approach (pcGEM) to improve the accuracy of enzyme usage predictions in *Chlamydomonas reinhardtii*. For proteomics, they used LC-MS coupled with QConCAT approach to generate absolute protein abundance data. Although I am not expert on the modeling parts of the work and the use of NIDLE and GECKO toolbox, the proteomics approach is sound and sufficient details on methodology and generated data/results are provided.

However, would be good if authors more specifically consider the followings when designing/describing QConCATS approach:

- Selected tryptic peptides from proteins of interest are reproducibly observed in standard LC-MS analyses and are unique to the protein of interest (i.e. the sequence is not shared with any other protein in the proteome of interest). Typically, QConCATS are designed around peptides that provide the greatest sensitivity.
- No methionines in selected peptides. Methionines can and will oxidize which then leads to poor quantification. (This one is already briefly mentioned in line 264, but a bit more clarification would be ideal).
- Flanking sequences after the K and R residues. Certain flanking residues can impact trypsin digestion efficiency. Proline is a good example to avoid, and acidic residues to a lesser extent (D and E).

If the QConCAT breaks some of these rules, I think one could no longer assume 1:1 stoichiometric ratio of the heavy tryptic peptides

Minor comments:

- The environmental parameters studied (Control, Dark, High Cell, High Salt, High Temp, No Shaking) are not well-justified. Are these conditions defined based on previous published

studies or based on preliminary experiments/data by the authors?

- What is the difference between SDP OE1 UVM4 and SDP OE2 UVM4 strains?
- Spell out XIC in line 267

Response Letter to Reviewers' Comments to

“Proteomics and constraint-based modelling reveal enzyme kinetic properties of *Chlamydomonas reinhardtii* on a genome scale”

The letter contains our point-by-point responses to all reviewers' comments. The original comments are displayed in regular font, while our responses are provided in bold green font.

Reviewer #1 (Remarks to the Author):

Arend et al. present a global study of kinetic parameters in the metabolism of the green algae *Chlamydomonas reinhardtii*. There is much interest in getting new insight into algae as they can contribute to greenhouse gas mitigation in terms of converting carbon dioxide to food, fuels and chemicals. In this study a comprehensive dataset of semi-absolute protein concentrations across 9 conditions, obtained by merging previous published data with additional new measurements, is made available. The culture conditions span autotrophic, mixotrophic and heterotrophic metabolism. This study uses a genome-scale metabolic model and physiology data to compute metabolic fluxes using a MILP approach (NIDDLE). Next, maximum apparent enzyme turnover numbers (k_{app}^{max} , which reflect the maximum capacity of an enzyme at in vivo conditions) are calculated assuming proportionality between the computed fluxes and the measured protein abundances. The authors highlight their large contribution to the extension of data on kinetic parameters for green algae's metabolism (from 85 parameters to 568 entries). The computed kinetic parameters provide a high number of entries in lipid metabolism and mitochondrial transport, which were not covered by the values in databases.

k_{app}^{max} values are compared to the enzyme turnover numbers matched by the GECKO method to each of the reactions of a protein-constrained version of the GEM, finding that GECKO values are systematically higher than the computed k_{app}^{max} , while the latter offer a catalogue of condition dependent catalytic rates. Their results support previous findings reporting that in vitro kinetic parameters do not match k_{app}^{max} values computed from metabolic fluxes and proteomics in eukaryote organisms. Different protein-constrained models are generated, based on the GECKO formalism, using kinetic parameters from different origins (computed k_{app}^{max} and literature k_{cat} assigned by GECKO). Such models are used to evaluate predictive performance on maximum growth rate and protein usage across several conditions. The authors conclude that the use of computed k_{app}^{max} for parameterization of protein-constrained models improves quality of phenotype predictions.

Evaluation

The manuscript is generally well written and concise. Overall, the study provides valuable results that contribute to extend the available knowledge of *Chlamydomonas reinhardtii* metabolism, using a computational approach, aided by generation of high-throughput data. One could have wished for a more in-depth analysis of the physiology of the organism using the model, i.e. novel biological insights, but in light of the large amount of work carried out this is not a key issue. Therefore, I consider that the quality and relevance of this work is suitable for publication in Nature Communications after the following major and minor points are addressed.

We thank the reviewer for the generally positive assessment. In the updated version, we highlighted the differences between biological conditions and metabolic pathways, thus fully addressing the constructive criticism by the reviewer.

Major points:

1.- Lines 163-167: Here the details for the comparison between GECKO k_{cat} s and computed k_{app}^{max} is detailed. Low correspondence of values is explained here just by a low Spearman correlation coefficient, which is a measure of how monotonic the relation between two distributions of values is. Generalized and Michaelis-Menten enzyme kinetic expressions clearly show that the capacity limit of enzymes (k_{cat}) is only one of several terms/factors affecting reaction rates, from this it is expected that *in vivo* capacities differ from those *in vitro* to an extent that is dictated by context dependent properties. This is in line with the sentence indicating that GECKO k_{cat} values are systematically higher than computed k_{app}^{max} . But to what extent? It is suggested to characterize the discrepancy between these two distributions in quantitative detail. For instance, statistics on ratios between \log_{10} values from the two distributions will inform of how many orders of magnitude are GECKO values higher than k_{app}^{max} , a relevant issue when thinking of kinetic parameters as constraints for metabolic flux predictions. Additionally, how large are those differences in enzymes grouped by metabolic context (subsystems)? With such an analysis, focusing on *C. reinhardtii* or taxon-specific values, other biological features can be explored, which may be of interest to an audience even broader than metabolic modelers, for instance, for instance pathway specialization of enzymatic regulation and control in *C. reinhardtii*.

We would like to thank the reviewer for the suggestion to systematically investigate the findings from our computational analysis. It has been shown repeatedly that, in *E. coli*, the k_{kapp}^{max} values obtained from aggregating experimental data from diverse conditions show relatively high correspondence with *in vitro* k_{cat} values^{1,2}. While this demonstrates that the applied approach results in condition-independent estimates of the maximum *in vivo* catalytic capacity, the obtained values in eukaryotes are not as well correlated to *in vitro* data^{2,3} --a result we also found to hold for the model green alga that we investigate in this study. To provide the reader with further information of the observed differences between k_{kapp}^{max} and k_{cat} values, as suggested, we investigated the \log_{10} -fold differences (LFD) between these values and added

- a boxplot of LFD grouped by metabolic subsystems (Macrosystem) (Fig. S5b). While nonparametric test could not identify significant differences in means of LFDs between metabolic subsystems, we observed differences in the variance quantified by the interquartile range. These differences are discussed in revised “Results and Discussion” section (lines 199-202).
- a boxplot of LFD for all enzymatic reactions in the intersect of NIDLE and GECKO and only for reactions with *C. reinhardtii*-specific k_{cat} (Fig. S5a) as well as a scatter plot of these values (Fig. S5c). We showed that Spearman correlation is higher when only endogenous k_{cat} measurements are taken into account. These insights are discussed in the “Results and Discussion” section (lines 197-199, 208-211).
- a boxplot of LFD between the condition-specific k_{app} (rather than the maximum over all conditions) and k_{cat} values (Fig 3b). Here the sample size was large enough to warrant the use of frequentist statistics. We found that the k_{app} values in heterotrophic growth depict significantly larger differences to the respective

literature k_{cat} when compared to stress-free mixotrophic conditions. These insights are discussed in the “Results and Discussion” section (lines 202-206) (see answer to point 4).

2.- Lines 173-174: Here the authors state that literature turnover numbers are a suboptimal source of parameters for pcGEMs. This is an issue that is hard to generalize, as pcGEMs may serve for different purposes, as any other GEM they are a comprehensive knowledgebase of an organism’s available information. For some applications predictive performance is important, in such cases the use of K_{app} is relevant for refining predictions (explored in the next section). However, literature turnover numbers are still an important source of parameters for non-model organisms for which exchange fluxes and proteomics data are scarcely available. Heuristic kinetic parameterization of pcGEMs has been used for quick generation of models for poorly characterized organisms and cell types, and have been helpful at providing general insights of their metabolism (Robinson et al. 2020; Lu et al. 2021).

We thank the reviewer for drawing attention to this generalization and for providing additional literature to facilitate a more detailed discussion of the scenarios in which k_{app}^{max} values are preferable to literature turnover numbers. We fully agree that in the scenarios mentioned by the reviewer, pcGEMs parameterized with literature turnover numbers can generate valuable insights. We have extended and rewritten the respective sections to account for a more nuanced discussion of this topic (lines 215-221).

3.- In the section “Parameterization of pcGEMs with the estimates of in vivo k_{app}^{max} values show improved enzyme usage prediction” performance comparisons between three models are performed. pcGEM with NIDLE parameters, a raw GECKO model and a calibrated GECKO model. Nevertheless, in the most recent publication of GECKO it is stated that the use of the “raw model” for quantitative simulation should be avoided, as this kind of model serves as a scaffold for proteomics data integration, therefore, only the calibrated GECKO model should be compared to the one with NIDLE-origin parameters. The sentence in lines 188-189, formulated as a finding, highlights what has been said previously and was the main motivation for the introduction of a parameter calibration procedure into GECKO2.0.

We agree with the reviewer that it has to be clearly stated that the raw model generated by GECKO is not meant to produce growth rate predictions and we have added such a statement in the revised manuscript (lines 243-247). However, we decided to show the performance of the raw model to provide an idea of the performance in photosynthetic eukaryotes where no data for the calibration procedure are available. This allows us to disentangle the effect of the calibration procedure on prediction accuracy from the effect of k_{kapp}^{max} parameterization, which is especially important when discussing the protein usage predictions (see answer to point 5, below). We amended the manuscript to stress that the results discussed in lines 188-189 of the initial submission are just a reconfirmation of previously reported observation in other taxa (now lines 247-249).

4.- Line 192: It was found that the fitted sigma factors differ across conditions, in GECKO sigma represents a lumped or average effect of condition-dependent factors over enzyme capacity. In contrast k_{app}^{max} values offer the opportunity of estimating such effects for each particular enzyme for which a *C. reinhardtii*, or related, K_{cat} value exists. In order to provide a more detailed analysis of this, please compare statistical metrics (e.g. median, mean,

coefficient of variance) for the distribution of ratios between k_{app}^{max} and K_{cats} across conditions versus the different sigma values.

We thank the reviewer for requesting this additional information, which led us to include the boxplot of LFD ($\log_{10} \frac{k_{app}}{k_{cat}}$) of condition-specific k_{app} values (Fig. 3b), already discussed in response to point 1, above. We would like to point out that the experimental data used in GECKO calibration procedure are from an independent chemostat study (Methods)⁴; thus, we have no proteomics or k_{app} data from these experiments. Yet, as noted in the manuscript, we were able to corroborate the conclusions drawn from the different sigma values with the suggested statistical analysis on LFD distributions in our independent proteomics experiments (lines 202-206, 254-256). Namely, enzymes in the heterotrophic condition show a significant lower LFD than those from mixotrophic non-stress conditions (Fig. 3b) (see answer to point 1, above).

In addition to the suggested condition-wise analysis, we also included more information on the reaction-wise differences in the form of a heatmap of clustered mean centred k_{app} values (Fig. 2a). This analysis showed that there is a small subset of reactions with highly varying k_{app} even for closely related conditions while the majority of reactions depict a variation of less than 2 on log10 scale (i.e. $< 10^2 \text{ s}^{-1}$). These new observations are introduced in the “Results and Discussion” section of the revised manuscript (lines 139-147).

5.- Lines 194-197: Growth rates are usually used as a calibration parameter for pcGEMs in GECKO, therefore, it is not directly a prediction from which predictive performance can be evaluated and compared to other approaches. It is suggested to use another prediction output (not used in calibration of the models) for comparison with the NIDLE parameters-based model.

The important concerns raised by the reviewer are the reason we decided to include the performance of the “raw model” in the assessment (Fig 4a) as mentioned in the answer to point 3, above. As the “raw model” does not rely on growth rates for calibration, it allows for a fair comparison of the effect of NIDLE parameters.

Furthermore, we agree that flux through the growth reaction, as a single predicted variable of the FBA LP, offers only a limited assessment of the effects of changing several hundred enzyme catalytic rate parameters. We find that the performance values based on the prediction of enzyme abundances are more meaningful as they are based on number of biological measurements that is even larger than the number of modulated parameters.

6.- Lines 215-217 and Figure 3B: Predictive performance of models is compared in terms of spearman correlation between simulated enzyme usages and measured protein abundances. This is a metric of correlation, however, it does not inform about how accurate predictions are. Please include other error metrics in analysis.

We followed this reviewer request and included a plot of Root Mean Squared Error (RMSE) values of the predicted enzyme usages (Fig. S6). This performance metric reconfirms the conclusions drawn from the Spearman correlation (k_{app}^{max} of NIDLE

improve model performance in all unseen scenarios). However, here again a higher error for all heterotrophic models is observed (see answers to points 1 and 4, above). These additional results are discussed in the revised “Results & Discussion” section (lines 278-287).

Additionally, computation of k_{app}^{max} values depend on input data. It is said that the tested condition was omitted from the calibration data for each case, however, if the training datasets are very similar to each other, then it is expected that predictions are good even when omitting the tested conditions. In figure 1 a PCA shows the differences at the protein expression level between conditions, additionally, an analysis of the different conditions in terms of both, used flux constraints, and simulated flux distributions is also required, in order to assess how different are the calibration data from the outputs of the tested condition. This will inform if observed predictive performance is somehow artificial or not.

We thank the reviewer for raising this important point. In addition to the flux bounds given in Table S5 we provide further analysis of the simulated flux distribution in the form of (i) a PCA of flux distributions (Fig S4e) and (ii) charts of the cumulative sum of the flux distributions from NIDLE (Fig S4f). The PCA plot showcases that flux solutions show a different pattern of separation in the principal component space than the k_{app} vectors. The cumulative sum charts point out strong difference in the absolute values of the different mixotrophic flux distributions. These insights are discussed in the “Results and Discussion” section of the revised manuscript (lines 152-155).

7.- It is recommended to offer more information about the condition dependency of computed k_{app}^{max} values, which is one of the major contributions of this paper. What is the variance and statistics of k_{app}^{max} across conditions, focusing on single enzymes. Obtention of genome-scale trends of condition dependency of catalytic rates for Eukaryotes might be a valuable resource for systems understanding of gene and protein regulation and control mechanisms.

We thank the reviewer for this constructive comment. We note that k_{app}^{max} for an enzyme denotes the maximum of all k_{app} values derived over different conditions and as such are condition-unspecific. This does not allow to compute the requested variances. Along the lines of the reviewer comment we included in the revised manuscript (i) an assessment of the difference in $k_{app}^{max}-k_{cat}$ LFD on a genome-scale, for each metabolic pathway (Macrosystem) included in the model (Fig. S5c; see answer to point 1, above) (ii) an assessment of the condition-wise $k_{app}-k_{cat}$ LFD (Fig 3b; see answers to points 1 and 4, above), and (iii) an assessment of the variance and Euclidian distance of k_{app} condition-wise profiles on a genomes scale (Fig S4d, see answer to point 4, above).

8.- k_{cat}^{\wedge} values offer an estimate of the physico-chemical limits of enzymes, for some applications it is important to know what are the limits for fine-tuning enzymes' performance, in this contexts k_{cats} become relevant. However, the general tone of the manuscript leaves the impression of dismissing k_{cats} measurement, it is suggested to review the article and make sure that a nuanced discussion about pros and cons of both the use of k_{cat}^{\wedge} and k_{app}^{max} . For what kind of applications one may be more useful and/or informative than the other in the context of pcGEM?

Prompted by the reviewer's suggestion we provided a more nuanced wording regarding the usage of k_{cat} measurements as well as the pros and cons of k_{app}^{max} vs. k_{cat} parameterizations, particularly for eukaryotes. See the updated Results and Discussion section (lines 188-189, 215-221). The usage of k_{app}^{max} vs. k_{cat} in model parameterization is explained in our answers to the points above.

Minor points:

8.- Line 36: "However, economic feasibility of photosynthetic bioreactors requires further optimization of desired biotechnological objectives", provide examples of the typical uses of "biotechnological objectives" that this algae has been used for, it will provide context for understanding the relevance of this study to non-modelers.

We extended the introduction to include the production of lipids⁵, pigments⁶ or hydrogen⁷ as examples for biotechnological objectives with practical relevance and cite references that exemplify the optimization of these targets in *C. reinhardtii* or other closely related Chlamydomonadales (lines 35-37).

9.- Line 44: New successful studies of metabolic engineering targets identification with protein constrained models are now available and published (e.g. targets for different products using a *B. subtilis* pcGEM, DOI: 10.3390/microorganisms11010178, and heme using a *S. cerevisiae* model, DOI: 10.1073/pnas.2108245119). The introduction focuses mostly on pcGEMs and kinetic parameters but a bit more of context is needed, so that a general audience can comprehend the relevance of this work and the extension of the field.

We thank the reviewer for the additional input which we included in the introduction section together with another successful pcGEM guided metabolic engineering study in *B. subtilis*⁸ (lines 43-45).

10.- Line 48: A single study (also by the authors) is cited as the only reference regarding computation of apparent enzyme catalytic rates. As the topic is a current active discussion with plenty of development in recent years, please add additional references, these parameters have previously been computed using flux balance analysis, resource balance analysis and ME-models for different organisms (*E. coli*, *S. cerevisiae* and *B. subtilis*). This will clarify the story of development in the field to the reader, especially to those not completely focused on FBA approaches, while also offering supporting evidence for the conclusions of this manuscript.

The respective section in the introduction (lines 49-55) was extended to give the requested additional details and primary references contained in the review cited in the previous version of the manuscript.

11.- Line 56-58: It is said that in order to make pcGEMs for green algae, a substantial increase of in vivo turnover numbers is needed. However, the current available tools (sMOMENT, ECMpy, GECKO) already enable integration of this kind of constraints to any standard metabolic network, rather, apparent in vivo kinetic parameters are needed for refining predictions of such models. It is suggested to edit the sentence accordingly.

The sentence the reviewer refers to has to be read in context with the previous sentence in the paragraph, in which we detail the very scarce availability of green algae specific

model parameters. We edited the sentence (lines 63-65) to emphasize that we discuss increasing the number of organism-specific kinetic parameters.

12.- Line 70: The term “unseen test conditions” is confusing in this context, as it requires further explanation to be understood, which is done below in the text. I suggest removing this from the sentence as this is an introduction, and the sentence already says that predictive accuracy of enzyme resource allocation was improved.

We removed this part of the text, as suggested, to avoid confusion.

13.- Line 73: Correct sentence to : “we employed a comprehensive, ...”

We have modified the text to address the noted issue.

14.- Line 80: The first time that the model iCre1355 is mentioned, however, the article referring to its original development is not cited here, which may confuse some readers and make them think that iCre1355 construction is also part of this manuscript’s work. Please cite.

The manuscript was adapted accordingly.

15.- Lines 82-84: The absolute number of enzymatic proteins between two different studies, that used different quantification methods and based on different organisms, is directly compared here, highlighting that the current study measured 100 more proteins than the previous. As proteomes differ drastically in size and composition, this kind of comparison does not reflect the “depth” of the studies. Rather, it is suggested to compare the extent of measured enzymatic proteins between different studies based on the percentage of measured enzymes out of the total number of known enzymes for a given organism (coverage), or enzymes in their models in this case.

We agree with the reviewer and rewrote the respective section to now compare the relative number of enzymes (lines 89-93) quantified in the studies.

16.- Figure 1 captions: The order of the subfigures in the description (a, b, c) does not correspond to the order in the figure.

This was fixed in the revised version of the manuscript.

17: Figure 1b: The total protein measured per sample is quantified in term of amol/cell. Then, these quantities are compared across conditions, showing notorious differences with no systematic patterns. What if in a given stress condition a cell needs to highly overexpress a few stress-coping proteins, which have been found to tend to be relatively light proteins (Doughty et al. 2020), while keeping its central carbon metabolism enzymes (which tend to be heavier and less likely to be significantly differentially expressed) mostly unchanged? This situation may lead to an scenario that shows increased number of total protein molecules, but may not reflect the fact that such changes are not significant in a genome-scale. It is suggested to change representation of this figure by comparing protein mass (pg/cell) across conditions and

reevaluate the argument that no systematic patterns of total protein expression were observed across conditions (written in lines 94-95).

We agree that plotting the enzyme mass instead of the amount is an important additional information on the proteomics data. Therefore, we included a new figure that shows the requested values for the different conditions (Fig S1). As a result, we were able to show that the ranking of conditions by total amount and mass differs clearly, whereas the subset of proteins annotated as enzymes in iCre1355 depicts very similar trends over the different conditions in both amount and mass. These insights are discussed in the revised “Results and Discussion” section (lines 102-105).

18.- Line 117: “which is the largest set of organism-specific k_{app}^{\wedge} estimates generated to date”. The statement highlights technical relevance, but not necessarily biological relevance, it is suggested to normalize the number of quantified proteins and or protein enzymes, according to the number of known proteins in *C. reinhardtii* and/or number of known enzymes. It may also be convenient to add a comparison on the number of studied conditions and strains, which may emphasize the scale of this study.

Since our ability to make accurate biological predictions scales with the number of *in vivo* parameters that are for a pcGEM, we followed the reviewer’s suggestion and now provide the percentage of all enzyme-catalyzed reactions for which we were able to estimate a k_{cat} (lines 125-127). We added this measure also for the two studies in *E. coli* and *S. cerevisiae* we compare our results to in the conclusion section (lines 293-297).

19.- Line 120: Specify the statistical test used for computing p-value and if it is one or two-tailed.

The manuscript was adapted accordingly.

20.- Line 121-123: Additional analysis of the core computed k_{app}^{\wedge} (group of $n=189$ obtained for the 9 conditions) I recommended, for instance in which sectors of metabolism or pathways do they participate? This kind of analysis will inform about the swaths of cell metabolism that are being explored and shed light on by this study, which is said to be a significant increase of knowledge for *C. reinhardtii*.

The requested analysis was added in the form of Figure S4b, that indicates the distribution of the core computed k_{app}^{max} values over the metabolic subsystems if iCre1355. We observe that many of these reactions belong to lipid and heterocycle synthesis and we discussed this finding in the revised “Results and Discussion” section (lines 132-133).

21.- Lines 123-124: It is mentioned that results, until this section, suggest that selection of a maximum k_{app}^{\wedge} across conditions, $k_{app}^{\wedge max}$, serve as a good approximation for *in vivo* turnover numbers. What is it meant by this? It is suggested to express this in terms of performance metrics. Additionally, so far in the manuscript, $k_{app}^{\wedge max}$ values have been compared between two different computation schemes (NIDLE and pFBA) this may inform about prediction precision, but not accuracy, and also in terms of coverage, however this is not informative of how good the approach is in terms of reflecting *in vivo* conditions.

We have updated this section of the results to indicate that good approximation *in vivo* turnover numbers and k_{app}^{max} is expected when the latter are estimated over a sufficient number of samples (in our case, from all experiments) that provide a range of fluxes (lines 133-136).

This subsection of the results is named “Improved coverage of k_{app}^{max} ...”, in agreement with the remaining parts of the reviewer’s comment.

22.- Line 126: “operating at the maximum catalytic rate...”, as this the same term that it is used to describe k_{cat} , it is suggested to be substitute by “maximum *in vivo* catalytic rate”.

The manuscript was adapted accordingly.

23.- Lines 138-139: Check writing in point C in figure 2 captions. Not clear.

Figure captions were corrected accordingly.

24.- Figure 2D captions: please indicate what $\rho = 0.17$ means here.

Figure captions do now state that Spearman’s ρ is indicated.

25.- Line 142: It is suggested to change “of decreasing quality” to “of decreasing stringency/specificity”, as evaluation of a match quality requires comparison to an *in vivo* known value, case by case. However, the phrasing could be kept if different correlation values, or error metrics between GECKO k_{cats} and computed k_{app}^{max} , are estimated for the different groups of parameters matched by GECKO.

The section was adapted to “decreasing stringency”, since this best captures the meaning of the color code.

26.- Line 152: same suggestion as point number 22.

The manuscript was adapted accordingly.

27.- Lines 159-160: Change “*in vitro* determined turnover numbers provide a rather poor proxy of *in vivo* turnover numbers” to “*in vitro* determined turnover numbers provide a rather poor proxy of *in vivo* turnover numbers computed from protein abundance and estimated fluxes” or something similar. Since *in vivo* turnover numbers cannot be directly measured, therefore, their exact value is not available.

The sentence was adapted to “rather poor proxy of *in vivo* turnover number estimates”

28.- Lines 212-213: Avoid using the term “enzyme usage coefficients”, as the word coefficient

is usually used for non-variable values. Instead, enzyme usages are state variables of protein constrained models.

The manuscript was adapted accordingly.

Reviewer #2 (Remarks to the Author):

In the manuscript by Arend et al., the authors employed a protein-constrained genome scale metabolic model approach (pcGEM) to improve the accuracy of enzyme usage predictions in *Chlamydomonas reinhardtii*. For proteomics, they used LC-MS coupled with QConCAT approach to generate absolute protein abundance data. Although I am not expert on the modeling parts of the work and the use of NIDLE and GECKO toolbox, the proteomics approach is sound and sufficient details on methodology and generated data/results are provided.

However, would be good if authors more specifically consider the followings when designing/describing QConCATS approach:

- Selected tryptic peptides from proteins of interest are reproducibly observed in standard LC-MS analyses and are unique to the protein of interest (i.e. the sequence is not shared with any other protein in the proteome of interest). Typically, QConCATS are designed around peptides that provide the greatest sensitivity.
- No methionines in selected peptides. Methionines can and will oxidize which then leads to poor quantification. (This one is already briefly mentioned in line 264, but a bit more clarification would be ideal).
- Flanking sequences after the K and R residues. Certain flanking residues can impact trypsin digestion efficiency. Proline is a good example to avoid, and acidic residues to a lesser extent (D and E).

If the QConCAT breaks some of these rules, I think one could no longer assume 1:1 stoichiometric ratio of the heavy tryptic peptides

The QConCAT proteins were designed following the comprehensive guidelines outlined in Hammel et al⁹. In essence, this approach focused exclusively on proteotypic peptides that had been consistently detected in previous LC-MS/MS studies. Furthermore, we incorporated the d::pPop algorithm¹⁰ to enhance our selection process using artificial intelligence. To improve the reliability of our peptide spectrum matching workflow strategy, we accounted for methionine oxidation and its potential impact on quantification. Additionally, we included three or more peptides per protein on each QConCAT, thereby increasing the probability of accurate quantification, even if one peptide had suboptimal quantification results. This information was added as subsection in the Methods part of the revised manuscript (lines 325-336)

Minor comments:

- The environmental parameters studied (Control, Dark, High Cell, High Salt, High Temp, No Shaking) are not well-justified. Are these conditions defined based on previous published studies or based on preliminary experiments/data by the authors?

The chosen environmental parameters were selected considering previous studies of *C. reinhardtii*. We aimed to select conditions where notable variations in protein abundances or physiology were previously observed or could be anticipated based on previous studies. The intention behind this selection was to encompass a diverse range of the proteome and metabolome landscape of *Chlamydomonas reinhardtii*, thus allowing us to stretch metabolism allowing us to obtain the estimated of k_{app}^{max} .

Reference Publications:

Dark/Anoxic:

Subramanian V, Dubini A, Astling DP, Laurens LM, Old WM, Grossman AR, Posewitz MC, Seibert M. Profiling *Chlamydomonas* metabolism under dark, anoxic H₂-producing conditions using a combined proteomic, transcriptomic, and metabolomic approach. *J Proteome Res.* 2014 Dec 5;13(12):5431-51. doi: 10.1021/pr500342j. Epub 2014 Oct 21. PMID: 25333711.

Dubini, A.; Mus, F.; Seibert, M.; Grossman, A. R.; Posewitz, M. C. Flexibility in anaerobic metabolism as revealed in a mutant of *Chlamydomonas reinhardtii* lacking hydrogenase activity *J. Biol. Chem.* 2009, 284 (11) 7201– 13

Mus, F.; Dubini, A.; Seibert, M.; Posewitz, M. C.; Grossman, A. R. Anaerobic acclimation in *Chlamydomonas reinhardtii*: anoxic gene expression, hydrogenase induction, and metabolic pathways *J. Biol. Chem.* 2007, 282 (35) 25475– 86

Terashima, M.; Specht, M.; Naumann, B.; Hippler, M. Characterizing the anaerobic response of *Chlamydomonas reinhardtii* by quantitative proteomics *Mol. Cell. Proteomics* 2010, 9 (7) 1514– 32

Zhang, Z., Tan, Y., Wang, W. et al. Efficient heterotrophic cultivation of *Chlamydomonas reinhardtii*. *J Appl Phycol* 31, 1545–1554 (2019). <https://doi.org/10.1007/s10811-018-1666-0>

High Salt:

Fal S, Aasfar A, Rabie R, Smouni A, Arroussi HE. Salt induced oxidative stress alters physiological, biochemical and metabolomic responses of green microalga *Chlamydomonas reinhardtii*. *Heliyon.* 2022 Jan 21;8(1):e08811. doi: 10.1016/j.heliyon.2022.e08811. PMID: 35118209; PMCID: PMC8792077.

Heat:

Mühlhaus T, Weiss J, Hemme D, Sommer F, Schroda M. Quantitative shotgun proteomics using a uniform ¹⁵N-labeled standard to monitor proteome dynamics in time course experiments reveals new insights into the heat stress response of *Chlamydomonas reinhardtii*. *Mol Cell Proteomics.* 2011 Sep;10(9):M110.004739. doi: 10.1074/mcp.M110.004739. Epub 2011 May 24. PMID: 21610104; PMCID: PMC3186191.

Zhang N, Mattoon EM, McHargue W, Venn B, Zimmer D, Pecani K, Jeong J, Anderson CM, Chen C, Berry JC, Xia M, Tzeng SC, Becker E, Pazouki L, Evans B, Cross F, Cheng J, Czymmek KJ, Schroda M, Mühlhaus T, Zhang R. Systems-wide analysis revealed shared and unique responses to moderate and acute high temperatures in the green alga *Chlamydomonas reinhardtii*. *Commun Biol.* 2022 May 13;5(1):460. doi: 10.1038/s42003-022-03359-z

- What is the difference between SDP OE1 UVM4 and SDP OE2 UVM4 strains?

The differences between both strains are described in detail in Hammel et al 2020. In short, both strains represent overexpression lines of the Sedoheptulose-1,7-Bisphosphatase using different constructs, while one encodes a 3xHA tag at the C-terminus, the other lacks any tags¹¹.

- Spell out XIC in line 267

The manuscript was adapted accordingly.

References

1. Davidi, D. *et al.* Global characterization of in vivo enzyme catalytic rates and their correspondence to in vitro kcat measurements. *PNAS* **113**, 3401–3406; 10.1073/pnas.1514240113 (2016).
2. Chen, Y. & Nielsen, J. In vitro turnover numbers do not reflect in vivo activities of yeast enzymes. *PNAS* **118**, e2108391118; 10.1073/pnas.2108391118 (2021).
3. Küken, A., Gennermann, K. & Nikoloski, Z. Characterization of maximal enzyme catalytic rates in central metabolism of Arabidopsis thaliana. *The Plant Journal* **103**, 2168–2177; 10.1111/tpj.14890 (2020).
4. Imam, S. *et al.* A refined genome-scale reconstruction of Chlamydomonas metabolism provides a platform for systems-level analyses. *The Plant journal : for cell and molecular biology* **84**, 1239–1256; 10.1111/tpj.13059 (2015).
5. Chang, R. L. *et al.* Metabolic network reconstruction of Chlamydomonas offers insight into light-driven algal metabolism. *Molecular systems biology* **7**, 518; 10.1038/msb.2011.52 (2011).
6. Flassig, R. J., Facht, M., Höffner, K., Barton, P. I. & Sundmacher, K. Dynamic flux balance modeling to increase the production of high-value compounds in green microalgae. *Biotechnology for biofuels* **9**, 165; 10.1186/s13068-016-0556-4 (2016).
7. Dal'Molin, C. G. d. O., Quek, L.-E., Palfreyman, R. W. & Nielsen, L. K. AlgaGEM--a genome-scale metabolic reconstruction of algae based on the Chlamydomonas reinhardtii genome. *BMC genomics* **12 Suppl 4**, S5; 10.1186/1471-2164-12-S4-S5 (2011).
8. Massaiu, I. *et al.* Integration of enzymatic data in Bacillus subtilis genome-scale metabolic model improves phenotype predictions and enables in silico design of poly-γ-glutamic acid production strains. *Microbial cell factories* **18**, 3; 10.1186/s12934-018-1052-2 (2019).
9. Hammel, A., Zimmer, D., Sommer, F., Mühlhaus, T. & Schroda, M. Absolute Quantification of Major Photosynthetic Protein Complexes in Chlamydomonas reinhardtii Using Quantification Concatamers (QconCATs). *Frontiers in plant science* **9**, 1265; 10.3389/fpls.2018.01265 (2018).
10. Zimmer, D., Schneider, K., Sommer, F., Schroda, M. & Mühlhaus, T. Artificial Intelligence Understands Peptide Observability and Assists With Absolute Protein Quantification. *Frontiers in plant science* **9**, 1559; 10.3389/fpls.2018.01559 (2018).
11. Hammel, A. *et al.* Overexpression of Sedoheptulose-1,7-Bisphosphatase Enhances Photosynthesis in Chlamydomonas reinhardtii and Has No Effect on the Abundance of Other Calvin-Benson Cycle Enzymes. *Frontiers in plant science* **11**, 868; 10.3389/fpls.2020.00868 (2020).

REVIEWERS' COMMENTS

Reviewer #1 (Remarks to the Author):

I think the authors have done a great job in revising the manuscript and I do not have any further comments.

Reviewer #2 (Remarks to the Author):

Dear authors, Thank you for addressing all my comments on proteomics aspect of the work and thoroughly explaining and adding any missing information. I am satisfied with how you addressed the previous comments.